# Fibroblast Activation Protein Inhibitor (FAPI)-Based Theranostics—Where We Are at and Where We Are Heading: A Systematic Review

**DOI:** 10.3390/ijms24043863

**Published:** 2023-02-15

**Authors:** Marko Magdi Abdou Sidrak, Maria Silvia De Feo, Ferdinando Corica, Joana Gorica, Miriam Conte, Luca Filippi, Orazio Schillaci, Giuseppe De Vincentis, Viviana Frantellizzi

**Affiliations:** 1Department of Radiological Sciences, Oncology and Anatomo-Pathology, Sapienza, University of Rome, 00161 Rome, Italy; 2Department of Nuclear Medicine, Santa Maria Goretti Hospital, 04100 Latina, Italy; 3Department of Biomedicine and Prevention, University Tor Vergata, 00133 Rome, Italy

**Keywords:** FAPI, theranostics, fibroblast activation, FDG, PET

## Abstract

Cancer is the leading cause of death around the globe, followed by heart disease and stroke, with the highest mortality to this day. We have reached great levels of understanding of how these various types of cancer operate at a cellular level and this has brought us to what we call “precision medicine” where every diagnostic examination and the therapeutic procedure is tailored to the patient. FAPI is among the new tracers that can be used to assess and treat many types of cancer. The aim of this review was to gather all the known literature on FAPI theranostics. A MEDLINE search was conducted on four web libraries, PUBMED, Cochrane, Scopus, and Web of Sciences. All of the available articles that included both diagnoses and therapy with FAPI tracers were collected and put through the CASP (Critical Appraisal Skills Programme) questionnaire for systematic reviewing. A total of 8 records were deemed suitable for CASP review, ranging from 2018 to November 2022. These studies were put through the CASP diagnostic checklist, in order to assess the goal of the study, diagnostic and reference tests, results, descriptions of the patient sample, and future applications. Sample sizes were heterogeneous, both for size as well as for tumor type. Only one author studied a single type of cancer with FAPI tracers. Progression of disease was the most common outcome, and no relevant collateral effects were noted. Although FAPI theranostics is still in its infancy and lacks solid grounds to be brought into clinical practice, it does not show any collateral effects that prohibit administration to patients, thus far, and has good tolerability profiles.

## 1. Introduction

Cancer is the leading cause of death around the globe, outranking stroke and coronary heart disease mortality rates in many countries [1]. The COVID-19 pandemic did not aid in the overall management of cancer patients, delaying screening, diagnosis, and treatment for these patients. The Ukraine invasion also played its role last year, as many refugees and other citizens will miss or delay their treatment or screening in the aftermath of this war [2]. Cancer diagnosis can be done through different imaging modalities, among which positron emission tomography (PET) plays a central role. It permits in vivo imaging in patients by injection of a radiopharmaceutical, made of a β+-emitting isotope and a carrier, a molecule with a known biodistribution in the organism. The annihilation of a β+ particle and an electron, results in the emission of γ-rays in opposite directions, which are detected by the scanner. PET can tell, within some degree of error, where a metabolic process is taking place. Lately, we have seen great improvement in these tracers, bringing us to theranostics, where we are able to image and conduct diagnosis at the same time [3,4,5,6]. 18Fluoro-2-deoxyglucose (FDG) is the most used tracer in PET for oncologic imaging. FDG becomes internalized by the cell through the glucose transporters (GLUTs). As FDG enters the cell, it is phosphorylated by the hexokinase enzyme and does not undergo any other interaction in the cells, where it is trapped. Malignant, proliferating cells have modified metabolic pathways, which are seen because FDG overexpression in relation to the surrounding tissue, thus the contrast between structures is high [7]. The physiopathology lies in the Warburg effect, described by Otto Warburg 100 years ago, where the malignant cells could meet their energy demand by aerobic glycolysis [8]. PET finds a central role in TNM staging, as far as N and M are concerned, whereas, in T staging, CT plays a more pivotal role [9,10,11]. Although FDG accounts for the majority of PET, we are slowly but surely shifting to a more specific molecular targeted imaging with more specific tracers. FAPI-PET is an example of this shift that we are already living. Many studies have already been conducted evaluating FAPI in many types of tumor, both in diagnosis and theranostics settings. The tumor microenvironment (TME) plays a fundamental role in understanding cancer biology. TME is mainly found in the extracellular matrix (ECM), which is made up of blood vessels, growth factors, cytokines, and fibroblasts. The latter’s role is mainly collagen production and regulation of homeostasis and inflammation of the surrounding cells. Some fibroblasts have contraction properties that make them a special subpopulation, called myofibroblasts, with smooth muscle characteristics. These cells express fibroblast activation protein (FAP), which is the main subject of this review [12]. Cancer-associated fibroblasts (CAFs) are the main component of the TME and are found in most solid tumors. They have either tumor-suppressive or promoting activity. Both normal fibroblasts and CAFs share a spindle shape, although CAFs exhibit slight differences in cytoplasm branches and nuclei under a light microscope. They show high ECM synthesis and remodeling, leading to fibrosis and cancer progression [13]. It has been proven that FAP inhibition leads to a decrease in tumor vascularization [14]. FAP belongs to the dipeptidyl peptidase (DPP) protein family, which includes DPP4, FAP, DDP8, and DDP9. Therefore, FAP functions as an endopeptidase enzyme for substrates with glycine–proline motifs. A gelatinase function was also identified for FAP [15]. The monomeric form, known as FAPα, is an inactive state which is then activated by dimerization into FAPα/FAPα or the heterodimer, FAPα/FAPβ. This protein plays a crucial role in the remodeling of the TME [16]. Stromal fibroblast studies actually date back to the nineties, as imaging was done through labelled monoclonal antibody F19. mAB-F19 was cultivated in hybridoma cells or lung cancer cells in mice [17,18]. It was first studied in 1994 in metastatic colorectal cancer with ^131^I-mABF19. The rational was that the stromal compartment in TME makes up almost 50% of the normal tissue and imaging 3 to 5 days after injection. All lesions with positive uptake were then found to be sites of metastasis on biopsy. No therapeutic effect was seen whatsoever, as the patients were already in a very late stage of disease.

FDG dominates the oncologic nuclear imaging field, but FAPI might be a good candidate to dethrone the reigning tracer, as many oncologic imaging studies showed that FAPI had a higher potential for detecting tumor lesions. Benefits are also seen in patient compliance as imaging is done 10 min after injection, and so fasting beforehand is required, as uptake is independent of glucose blood levels [19].

The purpose of this review was to collect all known literature on FAPI theranostics, a weapon targeting the tumor microenvironment.

## 2. Methods

### 2.1. Search Strategy and Study Selection

This systematic review was set up according to Preferred Reporting Items for Systematic Reviews and Meta-Analyses (PRISMA) guidelines. Publications gathered included papers from 2018 to November 2022. The research was conducted on Pubmed, Cochrane Library, Web of Sciences, and Scopus. The following keywords were used as research terms: “FAPI” or “FAP” and “theranostic” or theragnostic”. Clinical studies involving the use of radiobound FAP for both diagnosis and therapy that included at least two patients were gathered and assessed for inclusion.

### 2.2. Data Extraction and Methodological Quality Assessment

Study design, patient characteristics, year of publication, country and authors’ generalities were retrieved for the included clinical studies. The Clinical Appraisal Skills Program (CASP), a tool frequently used for the systematic review of diagnostic accuracy was used to evaluate clinical studies [20] (See Figure 1).

## 3. Results

All eight studies that were deemed suitable for the review went through the CASP signaling question for the CASP diagnostic checklist. All studies had the clear goal of assessing radioligand therapy (RLT) in patients who showed a marked uptake on FAPI-PET and or FAPI scintigraphy, making these patients then eligible for therapy. All patients received therapy and pre-therapy scans with the same ligand, with a few exceptions because of adjustments in the ligand formula for scarce bioretention that would make therapy non-effective. These cases encompassed three studies, and this topic is expanded in the discussion session. All patients were oncologic and selected after the failure of previous lines of therapy and therefore were submitted to compassionate use of the RLT approved by the respective ethic committees. No study showed a relevant collateral effect whatsoever and there was a decent safety profile overall. As most of the patients were terminal, Progressive Disease was the most frequent outcome, followed by Stable Disease according to PERCIST and RECIST 1.1 criteria. Nevertheless, pain medication was greatly reduced with a non-negligible improvement in quality of life for the patients who underwent treatment. Results were matching for all the assessed studies. Most papers were produced in Germany, followed by India and then Iran and Turkey. A total of 74 patients were evaluated, 22 of which had thyroid cancer, 16 breast, 11 pancreas, 6 sarcoma, 5 ovarian, 3 colon, 2 prostate, 1 lung, 1 cervical, 1 rectal, 1 neuroendocrine, 1 paragangliomal, 1 cholangiocarcinoma, 1 chordoma, 1 thymic and 1 round cell tumor (See Table 1, Table 2 and Table 3).

## 4. Discussion

### 4.1. FDG vs. FAPI: A Brief Overview

Although FAPI-PET has been studied in various tumors, it seems to find a particular role in the oncologic evaluation of gastric cancer. Nevertheless, this does not exclude FDG usage in gastric cancer, as coupling of these two tracers greatly increases sensitivity and specificity for distant lesions when compared to each single tracer alone. FAPI was still superior to FDG for primary lesions [39]. FAPI was also found to be better than FDG in detecting lymph node metastasis and peritoneal carcinomatosis, with higher mean and median SUV compared to FDG and with higher tumor to background and tumor to liver ratios. These additional findings are important for patient management as it upstages the lesion overall [40]. FAPI’s superiority to FDG is yet to be confirmed in lung cancer. The use of FAPI-PET in lung cancer is, to this day, very low and studies show discrepancies as to which tracer shows a higher uptake or higher tumor to blood ratio. Some authors report similar results for TBR for both FAPI and FDG with no significant SUV difference [41]. Lung fibrosis and interstitial lung disease are a challenge, as they may be comparable to tumor lesion uptake, although later imaging shows faster washout from the scarred lung than the lung tumor [42]. To this date there is no literature that focuses primarily on FAPI-PET/CT for prostate cancer, aside from case reports of FAPI-positive scans that were negative on PSMA scans [43,44]. Breast cancer might also get good use out of FAPI. The first FAPI application in humans was reported by Baum et al., using ^68^Ga-FAP-2286 and ^177^Lu-FAP-2286 alongside other cancer patients, and no adversities were reported. Another case report of metastatic breast cancer imaged with ^68^Ga-DOTA.SA.FAPI and treated with ^177^Lu-DOTA.SA.FAPI reported a beneficial outcome for the patients, as her pain medication was reduced overall [25,45]. Differentiated thyroid cancer (DTC) might need both tracers to accurately stage the disease, as FAPI shows higher uptake than FDG in lymphatic lesions and local recurrences of disease, whereas other sites seem to be more FDG avid, making the diagnostic performance for FDG comparable to that of FAPI PET [46]. Another important use of these tracers worth discussing is in bone metastasis evaluation. The most common tools to assess bone metabolism are bone scans with diphosphonates and FDG-PET, the first one assesses the bone reaction of the lesion, whereas the second assesses the glucose metabolism within the lesion [47,48]. In another review evaluating bone metastasis with FAPI and FDG PET/CT, sensitivity was always close to 100% for FAPI, a value reached only in one study by FDG. On the other hand, FDG performed better on specificity [49].

### 4.2. FAPI Non-Oncologic Uptake

FAPI uptake has been described in non-tumor lesions. Since fibroblast activation can remodel the tissue in fibrosis, inflammation and tissue healing, this may lead to tracer uptake in non-cancerous tissue. It is also worth noting that FAPI tracers have also been studied in non-oncologic diseases, such as IgG4-related disease [50], cardiac amyloidosis [51,52,53], risk of sudden cardiac death [54], tuberculosis [55], Crohn’s disease [56], Erdheim–Chester disease [57], arthritis and fibrosis. A slight uptake is also seen in the thyroid that may increase in cases of chronic thyroiditis and immune related thyroiditis [58,59]. Physiological activity is also seen transiting in the intestines [30]. Bone marrow also exhibits low physiological activity, which increases in bone degenerative diseases and bone fractures [60]. A common pitfall to beware of is related to reactive lymph nodes, as authors have reported 7.7% of lymph nodes to be FAPI positive, most commonly in mediastinum, neck, and axilla and inguinal regions, but still with low uptake compared to metastatic lymph nodes [61].

### 4.3. CASP Clinical Studies

In Figure 2 and Figure 3 we can see the FAPI tracers that were used in both and clinical and pre-clinical settings, with and without chelators, based on bioretention and tumor affinity.

Lindner’s group was the first to report in 2018 development of the FAPI tracer and then use it in imaging and therapeutic settings. FAPI-04 turned out to be the most promising because of good stability in human serum, high affinity for FAP, and high tumor uptake. Two patients with metastatic breast cancer were evaluated with ^68^Ga-FAPI-04 and then treated with ^90^Y-FAPI-04. No adverse reactions were reported, and the clinical response was seen as a significant reduction in pain medication. The authors had developed numerous FAPI ligands prior to therapy. The best performing were FAPI-04 and FAPI-13. Compared to existing FAPI-02 data, these two tracers had better accumulation in tumor lesions, mainly 24h after injection [27]. In 2020 the same group developed another FAPI tracer (FAPI-34) for imaging purposes. Biodistribution was assessed in animal samples. Human administration of FAPI-34 was done in two metastatic pancreatic and ovarian cancer patients who had already received a ^68^Ga-FAPI-46 scan and ^90^Y-FAPI-46 therapy. Follow-up was done with ^99m^Tc-FAP-34, which showed the same lesions both in SPECT and in PET. The first bound ligand, FAPI-19, had very high internalization rate (95%). Interestingly, follow-up cold (unlabeled) FAPI-19 addition in tumor cells showed suppression of ^99^Tc-FAPI-19, owing to high affinity and specificity of the compound. Out of the compared FAPI ligands (-19 -28, -29, -33, -34, -43), FAPI-34 had the lowest uptake in excretory organs and higher uptake in tumor lesion in mice. [28].

In 2021 Ballal et al. developed two FAPI tracers, ^177^Lu-DOTA.SA.FAPI and ^177^Lu-DOTAGA.(SA.FAPI)2, with the aim of using them in patients with high FAP expression confirmed with ^68^Ga-DOTA.SA.FAPI PET/CT. The tracers were administered to 10 patients (3 and 7, respectively) with a variety of metastatic tumors. ^177^Lu and ^68^Ga PET/CT lesions were concordant. The downsides of the study were, in the first place, the small patient sample, as well as the use of two tracers, further subdividing the patient samples; ^177^Lu-DOTA.SA.FAPI did not prove effective because of rapid clearance (within 48 h p.i.). Thus, the first group received one therapy cycle, whereas the second group received three cycles. In general, patients treated with ^177^Lu-DOTAGA.(SA.FAPI)2 reported a clinical response, whereas patients treated with ^177^Lu-DOTA.SA.FAPI relapsed after an initial improvement and two of them died [21]. Again in 2022, the Indian group published their work on radioiodine refractory differentiated thyroid cancer (RR-DTC). Fifteen patients who were positive on a ^68^Ga-DOTA.SA.FAPI scan received ^177^Lu-DOTAGA.(SA.FAPI)2. Four patients had partial response whereas three showed Stable Disease. Tireoglobuln was overall lower than baseline. Again, no relevant adverse events were noted [22].

Ferdinandus et al., in 2022, evaluated nine patients with heterogenous tumors using ^90^Y-FAPI-46. All patients underwent ^68^Ga-FAPI-46 PET/CT to assess tumor lesions with FAPI uptake, and renal function with ^99m^Tc-MAG3. No significant adverse effects linkable with certainty to the radioligand therapy were reported. Out of nine patients, only one received three cycles of therapy, two patients received two cycles of therapy, whereas the rest received just the first administration. This was reported as being due to patient death, clinical deterioration, or lack of focal ^90^Y-FAPI-46 uptake in a post-treatment ^90^Y-scan. Post-treatment evaluation in eight patients showed four Progressive and 4 Stable Diseases according to RECIST 1.1. Metabolic evaluation according to PERCIST 1.0 in seven patients showed five Progressive Metabolic Diseases, one Stable Metabolic Disease and one Partial Metabolic Response [23]. Another German group in 2022 administered ^177^Lu-FAP-2286 to cancer patients after a positive ^68^Ga-FAPI scan. Different cancer patients were included in the study in order to achieve a broader spectrum of cancers. All of them showed high tracer uptake and retention as well as no critical adverse effects whatsoever [25]. Still, all patients but two showed Progression Disease on RECIST 1.1 and the rest showed Stable Disease. Assadi et al., in 2021, enrolled 21 patients with various cancers after a positive diagnostic ^68^Ga-FAPI-46 PET/CT or ^177^Lu-FAPI-46 scintigraphy. Eighteen patients underwent ^177^Lu-FAPI-46 cycles. One-third of the patients had stable disease after therapy, with a slight improvement of symptoms, whereas the other had progressive disease. The toxicity profile was overall good, with only one patient experiencing adverse effects [24]. Kuyumcu et al., in 2021, also evaluated ^177^Lu-DOTA-FAPI-04 in patients with ^68^Ga-FAPI-04 uptake in different tumors, with the highest involvement being in bone, followed by lymph node and hepatic metastasis [26]. Other authors have tried different isotopes, such as ^153^Sm bound to FAPI, obtaining similar results to the abovementioned [64] and the use of ^188^Re as a theranostic couple with ^99m^Tc has been hypothesized [28]. In diagnostic settings, PET FAPI tracers have been shown to be superior to FDG in many types of cancer, including gastrointestinal carcinoma, lung adenocarcinoma, and breast and nasopharyngeal cancer [40,65,66,67,68]. Because of the low number of studies, we still cannot determine which FAPI ligand has the best distribution and retention properties, and validation of a gold standard is still to be made.

The most used FAPI tracers are FAPI-04 and FAPI-46 because of their high tumor uptake and high target-to-background ratio. These properties are due to the lipophilicity and the chemical modification of the linker region. In vitro studies showed fast clearance from cells for FAPI-04 after 4 hours, and very slow clearance for FAPI-21 and FAPI-46. FAPI-36 had a poorer image contrast, probably due to increased albumin binding in the blood [69]. FAPI-113 shares the same characteristic because of binding to albumin or other plasma proteins, although it did have the lowest half maximal effective concentration and highest tumor accumulation [27].

Another limitation lies in the low and very heterogeneous amount of cancer patients, with no clear benefits in specific tumors or specific mutations whatsoever. Although the therapeutic isotopes tested have mainly been the pure β emitters ^177^Lutetium and ^90^Yttrium, the clinical utilization of which has been well-established for the management of neuroendocrine tumors and liver cancer [70], α-therapy could find an even more fitting role in small and aggressive lesions because of its high linear energy transfer (LET) [71]. To this day FAPI α-therapy has only been reported in preclinical settings in mice with 225-Actinium (^225^Ac-FAPI-04) [72]. ^225^Ac is a pure α emitter that has seen application in prostate and neuroendocrine cancers [73,74,75]. This would require FAPI ligands with a long retention time in tumor lesions to make up for the long half-life of ^225^Ac (9920 days) as well as ^177^Lu (66,443 days). Furthermore, prospective studies in non-terminal patients with a relatively satisfying quality of life and low ECOG (Eastern Cooperative Oncology Group) score could bring attention to radioligand therapy in earlier lines of oncologic treatment.

## 5. Conclusions

FAPI theranostics is definitely at the vanguard of personalized medicine. Studies so far encourage this direction of targeting the tumor microenvironment both in diagnosis, in which it is proving to be superior to FDG, as well as in therapy, where no relevant adverse effect or bad tolerability profiles prohibit administration. Although these studies showed improvement of symptoms at best, it should be noted that the patients received the radioligand therapy for compassionate use, and we cannot exclude a better outcome if the treatment is administered in an earlier stage of the disease.

## Figures and Tables

**Figure 1 ijms-24-03863-f001:**
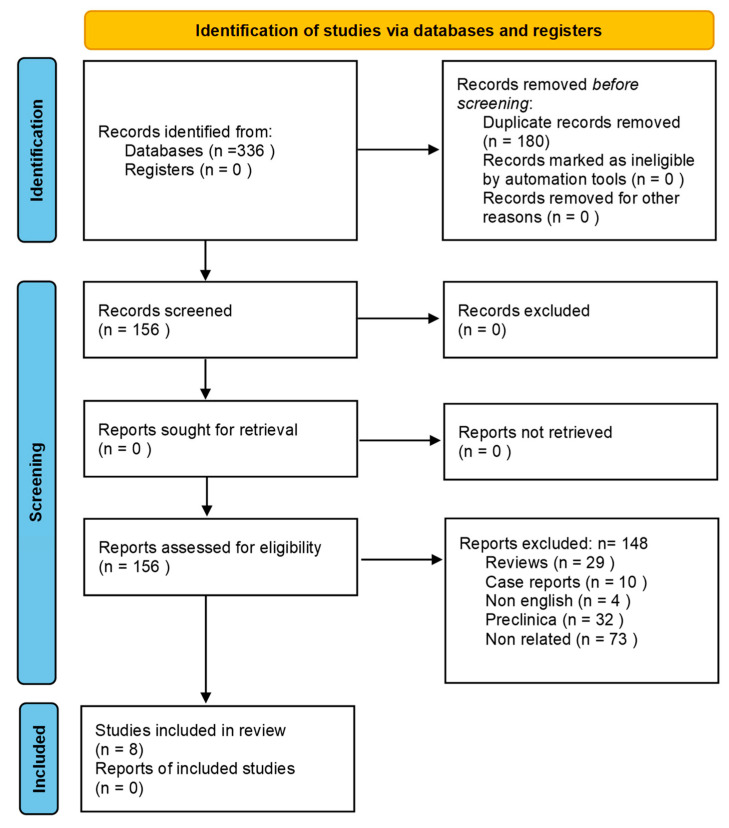
PRISMA flow-chart.

**Figure 2 ijms-24-03863-f002:**
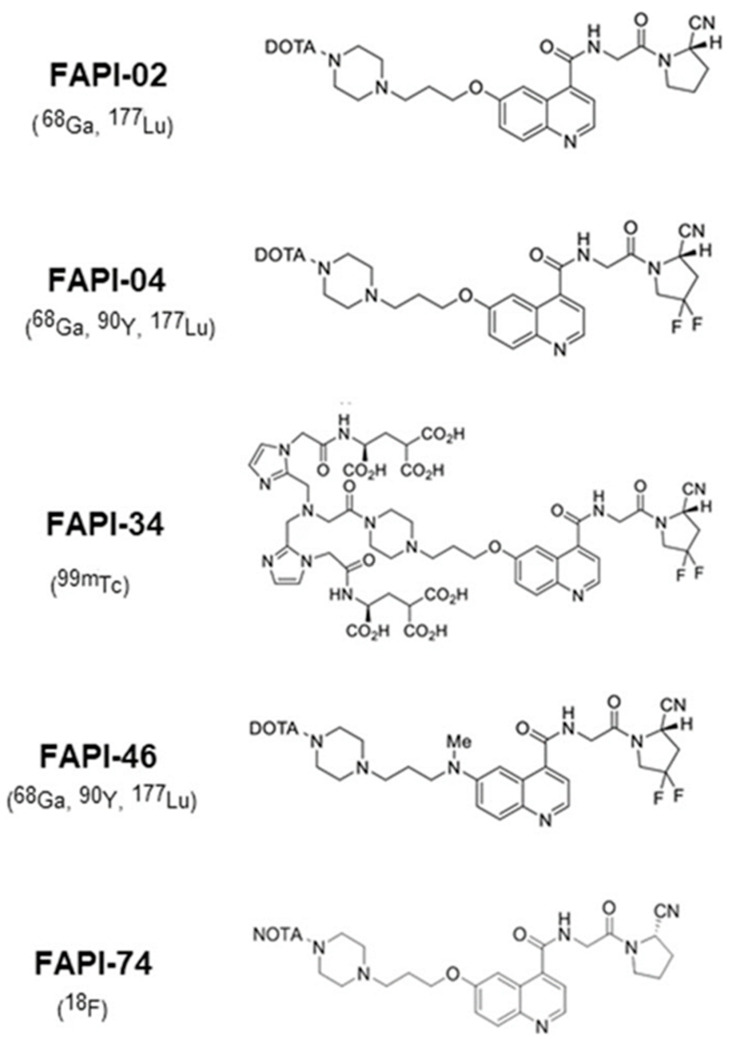
Chemical structures of FAPI precursors used in clinical application with nuclides in brackets [62]. Reprinted/Adapted from ref. [62].

**Figure 3 ijms-24-03863-f003:**
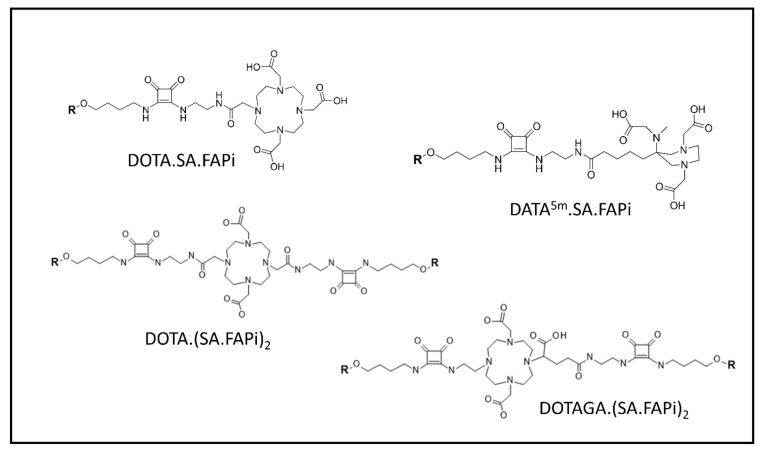
FAPI molecules developed by the Mainz research group [63]. Reprinted/Adapted from ref. [62].

**Table 1 ijms-24-03863-t001:** CASP diagnostic checklist.

	Ballal S. et al. 2021 [21]	Ballal S. et al. 2022 [22]	Ferdinandus J. et al. 2022 [23]	Assadi M et al. 2021 [24]	Baum RP et al. 2022 [25]	Kuyumcu S. et al. 2021 [26]	Lindner T. et al. 2018 [27]	Lindner T. et al. 2020 [28]
1. Was there a clear question for the study to address?	**✔**	**✔**	**✔**	**✔**	**✔**	**✔**	**✔**	**✔**
2. Was there a comparison with an appropriate reference standard?	**✔**	**✔**	**✔**	**✔**	**✔**	**✔**	**✔**	**✔**
3. Did all patients get the diagnostic test and reference standard?	**x**	**✔**	**✔**	**x**	**x**	**✔**	**✔**	**✔**
4. Could the results of the test have been influenced by the results of the reference standard?	**x**	**x**	**x**	**x**	**x**	**x**	**x**	**x**
5. Is the disease status of the tested population clearly described?	**✔**	**✔**	**✔**	**✔**	**✔**	**✔**	**✔**	**✔**
6. Were the methods for performing the test described in sufficient detail?	**✔**	**✔**	**✔**	**✔**	**✔**	**✔**	**✔**	**✔**
7. What are the results?	Both tracers were well tolerated. The first one had low tumor retention and was subbed out for the second one which had slower excretion and higher tumor retention and uptake.	No relevant adverse effect. Decrease in thyroglobulin was noted. 4 PR, 3 SD.	All tumors showed FAPI uptake. No major adverse events were noted. 7 PMD, 1 SMD, 1 PMR.	Toxicity profile was acceptable as only one patient experienced adverse effects. 12/18 patients who received the treatment had SD, whereas 6 had PD	These four types of cancer showed high FAP uptake. Decent safety profile with little retention. 9 PD, 2 SD	Bone involvement was noted with highest uptake. Critical organs absorbed dose was lower than other radioligands like ^177^Lu-PSMA-617 and ^177^Lu-DOTATATE	Imaging showed rapid uptake in tumor after 10 min from injection and high renal excretion with no retention.	Tumor lesions could be seen on both SPECT and PET.
8. How sure are we about the results, consequences and cost of alternatives performed?	All patients were considered terminal and received compassionate care through radioligand therapy. Stable Disease and Progressive Disease were to be expected. When Partial Response was achieved, palliative care medication was decreased.							
9. Can the results be applied to your patients or the population of interest?	**✔**	**✔**	**✔**	**✔**	**✔**	**✔**	**✔**	**✔**
10. Can the test be applied to your patients or population of interest?	**✔**	**✔**	**✔**	**✔**	**✔**	**✔**	**✔**	**✔**
11. Were all outcomes important to the individual or population considered?	**✔**	**✔**	**✔**	**✔**	**✔**	**✔**	**✔**	**✔**
12. What would be the impact of using this test on your patients/population?	As reported in these studies, palliative care medication was significantly reduced for a period of time with a slight non-negligible improvement in quality of life.							

SD = Stable Disease; PD = Progression Disease; SMD = Stable Metabolic Disease; PMD = Progressive Metabolic Disease; PMR = Partial Metabolic response.

**Table 2 ijms-24-03863-t002:** Characteristics of studies.

Author	Year of Publication	Country	Tracer	Population	Cancers
Baum et al. [25]	2022	Germany	^177^Lu-FAP-2286; ^68^Ga-FAPI-2286; ^68^Ga-FAPI-04	11 patients	5 pancreas; 4 breast; 1 rectum; 1 ovary.
Ferdinandus et al. [23]	2022	Germany	^90^Y-FAPI-46; ^68^Ga-FAPI-46	9 patients	3 pancreatic ductal adenocarcinoma; 4 sarcomas, 1 chordoma, 1 neuroendocrine tumor.
Lindner et al. [28]	2020	Germany	^99m^Tc-FAP-34; ^68^Ga-FAPI-46; ^90^Y-FAPI-46	2 patients	1 pancreas; 1 ovarian
Lindner et al. [27]	2018	Germany	^90^Y-FAPI-04; ^68^Ga-FAPI-04	2 patients	2 breast
Ballal et al. [21]	2021	India	^177^Lu-DOTA.SA.FAPI; ^177^Lu-DOTAGA.(SA.FAPI)_2_ ^68^Ga-DOTA.SA.FAPI	10 patients	5 thyroid; 4 breast; 1 paraganglioma
Ballal et al. [22]	2022	India	^68^Ga-DOTA.SA.FAPI; ^177^Lu-DOTAGA.(SA.FAPI)_2_	15 patients	15 thyroid cancers
Assadi et al. [24]	2021	Iran	^177^Lu-FAPI-46; ^68^Ga-FAPI-46	21 patients	2 ovarian cancer; 2 sarcomas, 3 colon cancer; 5 breast cancer; 2 pancreatic cancer; 2 prostate cancer; 1 cervical cancer; 1 lung cancer; 1 cholangiocarcinoma; 1 thyroid
Kuyumcu et al. [26]	2021	Turkey	^177^Lu-DOTA-FAPI-04; ^68^Ga-FAPI-04	4 patients	1 breast; 1 thymic carcinoma, 1 thyroid cancer, 1 ovarian cancer

**Table 3 ijms-24-03863-t003:** Direct comparison of detection rate of FAPI vs. FDG in various types of cancer.

Author	Year of Publication	Country	Tracer	Population	Cancers	Primary Lesion Detection Rate FAPI vs. FDG
Chen et al. [29]	2020	China	^68^Ga-DOTA-FAPI-04; ^18^F-FDG	75 patients	Heterogeneous types of cancer	98.2% vs. 82.1%
Pang et al. [30]	2020	China	^68^Ga-FAPI;	35 patients	GI tumor	100% vs. 53%
Jiang et al. [31]	2021	China	^68^Ga-FAPI-04; ^18^F-FDG	38 patients	Gastric cancer	100% vs. 82%
Zhao et al. [32]	2021	China	^68^Ga-FAPI; ^18^F-FDG	45 patients	Nasopharingeal cancer	100% vs. 97%
Qin et al. [33]	2021	China	^68^Ga-FAPI-04; ^18^F-FDG	15 patients	Nasopharingeal cancer	100% vs. 100%
Zhao et al. [34]	2021	China	^68^Ga-FAPI-04; ^18^F-FDG	46 patients	Peritoneal carcinomatosis	97.67% vs. 72.09%
Qin et al. [35]	2022	China	^68^Ga-DOTA-FAPI-04; ^18^F-FDG	21 patients	Gastric cancer	100% vs. 71.43%
Shi et al. [36]	2020	China	^68^Ga-FAPI; ^18^F-FDG	20 patients	Hepatic tumors	100% vs. 58.8%
Pang et al. [37]	2021	China	^68^Ga-FAPI; ^18^F-FDG	36 patients	Pancreatic cancer	100% vs. 73.1%
Kuten et al. [38]	2021	Israel	^68^Ga-FAPI-04; ^18^F-FDG	10 patients	Gastric cancer	100% vs. 50%

## Data Availability

Not applicable.

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
