# Peer review of "Fibroblast Activation Protein Inhibitor (FAPI)-Based Theranostics—Where We Are at and Where We Are Heading: A Systematic Review"

_ijms, 2023, doi:10.3390/ijms24043863_

Round 1
Reviewer 1 Report
The references listed are exhaustive.
like the listed table that lists therapy specific FAPI agents ; a similar listing may be made or added which is diagnostic specific- that deals with superiority of FAPI against conventional PETCT.
Author Response
We thank the Reviewer for the correct suggestion.
A FAPI PET diagnostic table showing recent studies directly comparing FAPI with FDG has been added
Reviewer 2 Report
FAPI is among the new tracers that can be used to assess and treat many types of cancers. The authors tried to gather all the known literature on FAPI theranostics, and put through the Critical Appraisal Skills Programme questionnaire for systematic reviewing. The authors made the conclusions that FAPI theranostics does not show any collateral effects that prohibit administration to patients, even it is still at the beginning and lacks solid grounds to come into clinical practice.
1. The Fibroblast activation protein (FAP) is a specific marker in activated fibroblasts. The FAP inhibitor (FAPI)-PET/CT-base imaging was also used for detecting cardiac injury or cardiac amyloidosis. Therefore, the title: “Fibroblast activation protein-based (FAPI) Theranostics, where we are at and where we are heading. A systematic review.” I would suggest to revise as follows: “Fibroblast activation protein inhibitor (FAPI)-based cancer theranostics, where we are at and where we are heading. A systematic review”.
2. Because of the low number of studies, the authors cannot determine which FAPI ligand has the best distribution and retention properties. However, FAPI as a promising theranostic probe, I would suggest authors discuss about the development of FAPI with improved tumor retention and antitumor efficacy in this review.
3. The section “4.3. CASP Clinical studies” is too long and should be improved. I would suggest to divide into two parts.
4. Define while first use and use abbreviated form of scientific names uniformly throughout the text. (e.g. FAPI vs FAPi)
5. What is the full name of “ECOG”?

Author Response
We want to thank the referee for his valuable suggestions:
The title has been changed
A short paragraph on FAPI chemistry has been added
Section 4.3 has been divided based on authors
Abbreviated form of scientific names have been corrected
ECOG full name has been added
Round 2
Reviewer 1 Report
the updated version appears fine to me